# Professional practice for COVID-19 risk reduction among health care workers: A cross-sectional study with matched case-control comparison

**Sarah Wilson[1]☯, Audrey Mouet[1], Camille Jeanne-Leroyer[1], France Borgey[2], Emmanuelle Odinet-Raulin[3], Xavier Humbert[4], Simon Le Hello[1,5]☯, Pascal Thibon[2]☯ \***

1 Service d'Hygiène Hospitalière et de Contrôle des Infections, Centre Hospitalo-Universitaire, Caen, Normandie, France, 2 Centre d'appui pour la Prévention des Infections Associées aux Soins, CPias Normandie, Centre Hospitalo-Universitaire, Caen, Normandie, France, 3 Agence Régionale de Santé Normandie, Caen, Normandie, France, 4 Normandie Univ, UNICAEN, UFR de Santé, Département de Médecine Générale, Caen, Normandie, France, 5 Groupe de Recherche sur l'Adaptation Microbienne (GRAM 2.0), Normandie Univ, UNICAEN, UNIROUEN, EA2656, Caen, Normandie, France

☯ These authors contributed equally to this work.
\* thibon-p@chu-caen.fr

**Data Availability Statement:** All relevant data are within the paper and its Supporting Information files.

## Abstract

### Background

Health care workers (HCWs) are particularly exposed to COVID-19 and therefore it is important to study preventive measures in this population.

### Aim

To investigate socio-demographic factors and professional practice associated with the risk of COVID-19 among HCWs in health establishments in Normandy, France.

### Methods

A cross-sectional and 3 case-control studies using bootstrap methods were conducted in order to explore the possible risk factors that lead to SARS-CoV2 transmission within HCWs. Case-control studies focused on risk factors associated with (a) care of COVID-19 patients, (b) care of non COVID-19 patients and (c) contacts between colleagues.

### Participants

2,058 respondents, respectively 1,363 (66.2%) and 695 (33.8%) in medical and medico-social establishments, including HCW with and without contact with patients.

### Results

301 participants (14.6%) reported having been infected by SARS-CoV2. When caring for COVID-19 patients, HCWs who declared wearing respirators, either for all patient care (ORa 0.39; 95% CI: 0.29–0.51) or only when exposed to aerosol-generating procedures

**Funding:** The author(s) received no specific funding for this work.

**Competing interests:** The authors have declared that no competing interests exist.

(ORa 0.56; 95% CI: 0.43–0.70), had a lower risk of infection compared with HCWs who declared wearing mainly surgical masks. During care of non COVID-19 patients, wearing mainly a respirator was associated with a higher risk of infection (ORa 1.84; 95% CI: 1.06–3.37). An increased risk was also found for HCWs who changed uniform in workplace changing rooms (ORa 1.93; 95% CI: 1.63–2.29).

## Conclusion

Correct use of PPE adapted to the situation and risk level is essential in protecting HCWs against infection.

## Introduction

Coronavirus disease 2019 (COVID-19) was declared a pandemic by the World Health Organization on March 11th 2020. There have since been approximately 216,000,000 cases and over 4,500,000 deaths worldwide [1]. Healthcare personnel are particularly vulnerable to infection given their exposure to the virus [2]. Between March 2020 and May 2021, 85,137 HCWs have been declared infected by the SARS-CoV2 virus in France, of which there have been 19 deaths. Within the infected personnel, 69% worked in clinical areas. The professions with the highest amount of infections were nurses (24% of cases) and nursing assistants (21%) [3].

SARS-CoV2 can be spread by respiratory droplets and fomite contact, as well as airborne transmission in specific circumstances [4,5]. However most transmissions occur during close face-to-face contact via respiratory droplets. The virus can be transmitted by presymptomatic, asymptomatic and symptomatic carriers [6,7]. Protection of HCWs is a key method for controlling the spread of the virus within health establishments, as vaccination does not provide complete protection against onward transmission. [8]. Guidelines recommend that when in contact with COVID-19 patients, and in addition to hand hygiene, HCWs protect themselves with personal protective equipment (PPE), namely surgical masks for standard care and respirators during aerosol-generating procedures, gowns and protective goggles [9,10]. Studies have suggested that SARS-CoV2 spreads not only between patients and from patients to HCWs, but also between infected HCWs, for example during breaks [7,11,12].

This study investigated sociodemographic factors, behavioral factors and professional practice associated with the risk of COVID-19 infection in healthcare workers. Secondary aims were to describe the circumstances of infection declared by the respondents, and the protective measures applied by healthcare professionals working in clinical areas, as well as during contacts with other colleagues.

## Methods

### Study design

A cross-sectional and three case-control matched studies were performed, based on an anonymous online questionnaire.

### Participants

Healthcare personnel (medical and paramedical professionals, as well as personnel from laboratories, hospital pharmacies and administration) working in health establishments (hospitals, clinics, rehabilitation and recuperation care facilities and establishments specializing in

psychiatry), nursing homes and establishments for handicapped children and adults in Normandy, France, were invited to participate in the study.

## Location

The study was conducted in Normandy, a region located in Northwestern France comprising of 6 departments (Calvados, Eure, Manche, Orne, Seine-Maritime), populated by 3,300,000 inhabitants [13], and with around 90,000 HCWs working in 197 health establishments, 522 establishments for handicapped adults and children, and 348 nursing homes [14].

## Rights and ethics

The study was approved by the local ethics committee for health research of Caen university hospital (ID 2293) on March 24th 2021. Participants received detailed information on the objectives of the study. Written agreement to use the anonymous data collected via the questionnaire was obtained for all participants, and they were informed of the possibility of the withdrawal of their data at any time, according to European regulation (27th April 2016).

## Period and data acquisition

The online questionnaire was available from 29th March 2021 to 30th June 2021. Healthcare personnel working in hospitals were invited to participate in the study by hospital management, who relayed the online version of the questionnaire. Healthcare workers from the medico-social sector were contacted by their management following an invitation to participate in the study via an email from the Regional Health Agency. The online questionnaire covered socio-demographic characteristics of HCWs (age, sex and profession), workplace, history of COVID-19 infection with date of infection (confirmed by a positive SARS-CoV2 PCR or antigenic test) as well as suspected exposures leading to COVID-19 infection, COVID-19 vaccination status, and personal preventative equipment and other barrier measures applied at work. For respondents who reported having been infected by SARS-CoV2, the personal preventative equipment and barrier measures were those applied at work during the ten days preceding infection symptoms (or testing in case of asymptomatic infection). For respondents with no history of COVID-19 infection, these measures were those applied at the time of filling in the questionnaire, and participants were asked if these practices had changed since September 2020.

## Cases and controls definition

Three case-control analyses were led, the first one describing measures applied during the care of COVID-19 patients, the second one those applied during the care of non-COVID-19 patients and the third one describing contacts with colleagues. Cases were defined as healthcare personnel who declared having had a COVID-19 infection (confirmed by a positive SARS-CoV2 PCR or antigenic test) which they reported as having been acquired in the workplace. Controls were healthcare personnel who declared no known history of COVID-19 infection over the study period and who declared no modifications of the personal preventative measures they applied since September 2020. Cases and controls were matched by sector of activity (health establishment or medico-social establishment) and by profession, with 4 controls for 1 case.

The study period for the case-control studies was defined as the period between the 1st September 2020 and the 31st January 2021, corresponding to the second wave of the COVID-19 pandemic in France, when recommendations for barrier measures had been issued [10], and

PPE was widely available. The cut-off point was chosen in order to study the effects of sociode-mographic factors, behavioral factors and professional practice before wide-spread vaccination of HCWs that started in January 2021.

## Statistical analysis

Qualitative variables were described with their effectives and percentages, and compared using the Chi-squared test. For each case-control study, the association between exposures and COVID-19 infection was measured by computing odds-ratios (OR) with univariate conditional logistic regression analysis, to take into account the matching of cases and controls on sector of activity and profession. As selecting a single set of controls could have led to an incorrect mea-surement of association due to random variations of ORs, we used a bootstrap method to per-form 1,000 random samplings of controls, with replacement. We then computed the mean ORs and their 95% confidence intervals (CI) (i.e. the 2.5% and 97.5% quantiles of the distribution of the 1,000 ORs) for each exposure. All variables significantly associated with risk of infection in the initial analysis were included in the multivariable analysis after testing for absence of collin-earity between variables. Analyses were performed using multivariate conditional logistic regression analysis and the same bootstrap method, allowing to calculate adjusted ORs (ORa).

$P < .05$ was considered significant. The analyses were performed in R version 4.1.1 (R Development Core Team).

## Results

The cross sectional study included 2,058 complete responses filled in by HCWs. The majority of participants worked in medical establishments (1,363, 66.2%) and 695 (33.8%) worked in med-ico-social establishments. A large proportion of participants (791, 38.0%) worked in non-medi-cal areas, regardless of the type of establishment. Nurses and nursing assistants represented 31.3% of respondents (N = 645), and doctors 10.3% (N = 212). The percentages of nurses, nurs-ing assistants and doctors were higher within respondents in health establishments than in medico-social establishments ($p<10^{-3}$). Most of the participants were women (N = 1,680, 81.6%), and their age was predominantly between 30 and 49 (N = 1,215, 59.0% of respondents) (**Table 1**). Within health establishments, most of the HCWs worked in general hospitals or Uni-versity Hospital Centers (N = 1,233, 90.5%), 5.0% (N = 68) worked in psychiatry and 2.0% (N = 27) in rehabilitation and recuperation care facilities. Nursing homes represented the work-place for a large part of HCWs in medico-social establishments (N = 267, 38.4%).

At the time of filling in the questionnaire, over two thirds of respondents (N = 1,443, 70.1%) had received at least one dose of the COVID-19 vaccine, and 11.0% (N = 226) did not wish to be vaccinated.

There were 301 participants with history of COVID-19 (14.6%). The percentage of respon-dents with history of COVID-19 was similar in health establishments and in medico-social establishments, but differed according to the profession ($p<10^{-3}$): 22.3% (57/256) for nursing assistants, 17.2% (67/389) for nurses, 16.1% (34/212) for doctors, 13.1% (55/420) for other HCWs in contact with patients, and 11.3% (88/781) for HCWs not in contact with patients. Most respondents with history of COVID-19 declared symptoms associated with the infection (N = 261, 86.7%). Ten respondents were hospitalized with COVID-19, and one participant needed treatment in an intensive care unit. The majority of participants who had had COVID-19 reported the possible contamination source as being at their workplace (N = 171, 56.8%), of which 67.8% (N = 116) reported a contamination due to contacts with a COVID-19 positive patient and 32.2% (N = 55) due to contacts with a COVID-19 positive colleague. Approxi-mately a quarter of respondents (N = 75, 24.9%) reported having been contaminated outside

**Table 1. Participants' characteristics, globally, and by sector of activity.**

| Characteristics | Health establishments N (%) | Medico-social establishments N (%) | Total N (%) |
|---|---|---|---|
| Professions | | | |
| Nurses | 334 (24.5) | 55 (7.9) | 389 (18.9) |
| Nursing assistants | 182 (13.4) | 74 (10.7) | 256 (12.4) |
| Doctors | 187 (13.7) | 25 (3.6) | 212 (10.3) |
| Other HCWs in contact with patients | 198 (14.5) | 222 (31.9) | 420 (20.4) |
| Other HCWs not in contact with patients | 462 (33.9) | 319 (45.9) | 781 (38.0) |
| Age (years) | | | |
| <30 | 198 (14.5) | 97 (14.0) | 295 (14.3) |
| 30–39 | 389 (28.5) | 184 (26.5) | 573 (27.8) |
| 40–49 | 416 (30.5) | 226 (32.5) | 642 (31.2) |
| > = 50 | 360 (26.4) | 188 (27.1) | 548 (26.6) |
| Sex | | | |
| Female | 1,110 (81.4) | 570 (82.0) | 1,680 (81.6) |
| Male | 253 (18.6) | 125 (18.0) | 378 (18.4) |
| Vaccination status | | | |
| Vaccinated, 1 dose | 481 (35.3) | 194 (27.9) | 675 (32.8) |
| Vaccinated, 2 doses | 549 (40.3) | 219 (31.5) | 768 (37.3) |
| Non-vaccinated | 333 (24.4) | 282 (40.6) | 615 (29.9) |
| Amongst the non-vaccinated | | | |
| Wish to be vaccinated | 140 (42.1) | 133 (47.2) | 273 (44.4) |
| Do not wish to be vaccinated | 123 (36.9) | 103 (36.5) | 226 (36.7) |
| Do not know if they wish to be vaccinated | 70 (21.0) | 46 (16.3) | 116 (18.9) |
| History of COVID-19 | 196 (14.4) | 105 (15.1) | 301 (14.6) |

HCW: Healthcare worker.

of the workplace and 18.3% (N = 55) of respondents didn't know how they were infected. In 2020, most cases occurred in March, April, September, October, November and December. In 2021, most cases occurred at the beginning of the year (January, February and March).

**Table 2** presents the results of the case-control study performed among HCWs caring for COVID-19 patients. When compared with mainly use of surgical masks, the use of respirators during aerosol-generating procedures (ORa 0.56; 95% CI: 0.43–0.70) and the use of respirators for all care (ORa 0.39; 95% CI: 0.29–0.51) were both associated with a decreased risk of infection. Wearing a hair cap (ORa 0.78; 95% CI: 0.63–0.98) was also associated with a decreased risk of infection. Use of face shields or protective goggles, gowns, protective overshoes and use of gloves for all types of patient care, as well as regular airing of patients' or residents' rooms were not associated with risk of infection.

**Table 3** describes the results of the case-control study performed among HCWs not caring for COVID-19 patients. Wearing mainly a respirator (ORa 1.84; 95% CI: 1.06–3.37) was found to be associated with a higher risk of infection when compared to mainly use of surgical masks. Use of a face shield or protective goggles (ORa 3.10; 95% CI: 1.81–5.58) and use of gloves for all types of patient care (ORa 1.36; 95% CI: 1.10–1.67) were also associated with a higher risk of infection. No association with infection was found for use of gowns and plastic aprons, protective hair caps and overshoes, and for regular airing of patients/residents' rooms.

**Table 4** describes the risk of infection associated with contacts between colleagues and airing of communal areas. No association with infection was found for eating at the workplace canteen, taking breaks with other colleagues, and airing of communal areas. Changing of outfit

**Table 2. Case-control study 1: Exposures associated with risk of COVID-19 among HCWs caring for COVID-19 patients.**

| Characteristics | Cases* (n = 70) | Controls* (n = 280) | OR [95% CI] | ORa [95% CI] |
|---|---|---|---|---|
| Age (years) | | | | |
| < 30 | 14 (20.0) | 55 (19.6) | 1 (ref) | 1 (ref) |
| 30–39 | 22 (31.4) | 84 (28.9) | 1.06 [0.87–1.30] | 1.16 [0.93–1.44] |
| 40–49 | 17 (24.3) | 85 (30.4) | 0.91 [0.73–1.10] | 0.97 [0.76–1.20] |
| ≥ 50 | 17 (24.3) | 59 (21.1) | **1.34 [1.05–1.67]** | 1.26 [0.97–1.61] |
| Sex | | | | |
| Female | 61 (87.1) | 239 (85.4) | 1 (ref) | |
| Male | 9 (12.9) | 41 (14.6) | 0.85 [0.66–1.08] | |
| **Preventative measures (PPE wearing and other)** | | | | |
| Handrubbing with alcohol based handrub before and after patient care | | | | |
| Never/rarely | 0 (0) | 8 (2.9) | - | |
| Regularly/always | 70 (100) | 272 (97.1) | - | |
| Type of mask used | | | | |
| Mainly surgical masks | 22 (31.4) | 55 (19.6) | 1 (ref) | 1 (ref) |
| Surgical masks + respirators during aerosol-generating procedures | 35 (50.0) | 148 (52.9) | **0.54 [0.43–0.65]** | **0.56 [0.43–0.70]** |
| Mainly respirators | 13 (18.6) | 77 (27.5) | **0.38 [0.29–0.46]** | **0.39 [0.29–0.51]** |
| Face shield or protective goggles | | | | |
| Never/rarely | 30 (42.9) | 106 (37.9) | 1 (ref) | 1 (ref) |
| Regularly/always | 40 (57.1) | 74 (62.1) | **0.82 [0.70–0.94]** | 1.27 [0.99–1.55] |
| Disposable gown and plastic apron when needed | | | | |
| Never/rarely | 13 (18.6) | 37 (13.2) | 1 (ref) | 1 (ref) |
| Regularly/always | 57 (81.4) | 243 (86.8) | **0.64 [0.50–0.79]** | 0.96 [0.70–1.26] |
| Gloves, for all types of patient care | | | | |
| Never/rarely | 12 (17.1) | 48 (17.1) | 1 (ref) | |
| Regularly/always | 58 (82.9) | 232 (82.9) | 0.99 [0.78–1.22] | |
| Protective hair cap | | | | |
| Never/rarely | 33 (47.1) | 106 (37.9) | 1 (ref) | 1 (ref) |
| Regularly/always | 37 (52.9) | 174 (62.1) | **0.67 [0.58–0.78]** | **0.78 [0.63–0.98]** |
| Protective overshoes | | | | |
| Never/rarely | 56 (80.0) | 218 (77.9) | 1 (ref) | |
| Regularly/always | 14 (20.0) | 62 (22.1) | 0.88 [0.74–1.05] | |
| Regular airing of patients/residents' rooms | | | | |
| Never/rarely | 24 (34.3) | 91 (32.5) | 1 (ref) | |
| Regularly/always | 46 (65.7) | 189 (67.5) | 0.91 [0.76–1.07] | |

OR: Odds ratio. ORa: Adjusted odds ratio.

* Cases and controls were matched on workplace and occupation.

in the workplace changing rooms was associated with a higher risk of infection (ORa 1.93; 95% CI: 1.63–2.29). Participation in professional meetings was associated with a decreased risk (ORa 0.72; 95% CI: 0.60–0.84).

## Discussion

This study covered a wide range of health establishments and professions and focused on a period during which PPE was available and before widespread vaccination of HCWs in France. We found that when caring for COVID-19 patients, HCWs who declared using respirators, either for all patient care or only when exposed to aerosol-generating procedures, had a lower

**Table 3. Case-control study 2: Exposures associated with risk of COVID-19 among HCWs taking care of non COVID-19 positive patients.**

| Characteristics | Cases* (n = 84) | Controls* (n = 336) | OR [95% CI] | ORa [95% CI] |
|---|---|---|---|---|
| Age (years) | | | | |
| < 30 | 17 (20.2) | 66 (19.6) | 1 (ref) | |
| 30–39 | 25 (29.8) | 103 (30.7) | 0.95 [0.75–1.19] | |
| 40–49 | 23 (27.4) | 103 (30.7) | 0.94 [0.73–1.20] | |
| ≥ 50 | 19 (22.6) | 64 (19.0) | 1.05 [0.80–1.36] | |
| Sex | | | | |
| Women | 72 (85.7) | 284 (84.5) | 1 (ref) | |
| Men | 12 (14.3) | 52 (15.5) | 0.91 [0.72–1.15] | |
| **Preventative measures (PPE wearing and other)** | | | | |
| Handrubbing with alcohol based handrub before and after patient care | | | | |
| Never/rarely | 0 (0) | 17 (5.1) | - | |
| Regularly/always | 84 (100) | 319 (94.9) | - | |
| Type of mask used | | | | |
| Mainly surgical masks | 60 (71.4) | 245 (72.9) | 1 (ref) | 1 (ref) |
| Surgical masks + respirators during aerosol-generating procedures | 19 (22.6) | 78 (23.2) | 1.01 [0.83–1.21] | 0.81 [0.66–1.00] |
| Mainly respirators | 5 (6.0) | 13 (3.9) | **2.36 [1.45–4.00]** | **1.84 [1.06–3.37]** |
| Face shield or protective goggles | | | | |
| Never/rarely | 76 (90.5) | 327 (97.3) | 1 (ref) | 1 (ref) |
| Regularly/always | 8 (9.5) | 9 (2.7) | **3.78 [2.34–9.97]** | **3.10 [1.81–5.58]** |
| Disposable gown and plastic apron when needed | | | | |
| Never/rarely | 62 (73.8) | 273 (81.2) | 1 (ref) | 1 (ref) |
| Regularly/always | 22 (26.2) | 63 (18.8) | **1.59 [1.29–1.96]** | 1.22 [0.93–1.57] |
| Gloves, for all types of patient care | | | | |
| Never/rarely | 34 (40.5) | 167 (49.7) | 1 (ref) | 1 (ref) |
| Regularly/always | 50 (59.5) | 169 (50.3) | **1.56 [1.29–1.88]** | **1.36 [1.10–1.68]** |
| Protective hair cap | | | | |
| Never/rarely | 79 (94.0) | 321 (95.5) | 1 (ref) | |
| Regularly/always | 5 (6.0) | 15 (4.5) | 1.33 [0.91–2.06] | |
| Protective overshoes | | | | |
| Never/rarely | 83 (98.8) | 331 (98.5) | 1 (ref) | |
| Regularly/always | 1 (1.2) | 5 (1.5) | 0.73 [0.40–1.40] | |
| Regular airing of patients/residents' rooms | | | | |
| Never/rarely | 28 (33.3) | 124 (36.9) | 1 (ref) | |
| Regularly/always | 56 (66.7) | 212 (63.1) | 1.19 [0.99–1.42] | |

OR: Odds ratio. ORa: Adjusted odds ratio.

* Cases and controls were matched on workplace and occupation.

risk of infection compared to HCWs who declared using mainly surgical masks. On the contrary, when caring for non COVID-19 patients, wearing a respirator compared to a surgical mask was found to be a risk factor of infection. Numerous studies have described the transmission of SARS-CoV2, and masks and respirators are the key elements of PPE when caring for COVID-19 positive patients [5,15]. In other situations, the discomfort of the equipment leading to HCWs touching the respirator to adjust it, therefore contaminating their hands could explain our findings. Respirators also need to be well fitting, and a badly fitting respirator could lead to a false sense of security by not providing a sufficient level of protection [16]. Violante et al. published a systematic review of scientific literature on the protective efficacy of

**Table 4. Case-control study 3: Contacts between colleagues and airing of communal areas and risk of COVID-19 (all professions included).**

| Characteristics | Cases* (n = 109) | Controls* (n = 436) | OR [95% CI] | ORa [95% CI] |
|---|---|---|---|---|
| Age (years) | | | | |
| < 30 | 19 (17.4) | 69 (15.8) | 1 (ref) | |
| 30–39 | 32 (29.4) | 128 (29.4) | 1.02 [0.85–1.28] | |
| 40–49 | 31 (28.4) | 128 (29.4) | 1.03 [0.74–1.26] | |
| ≥ 50 | 27 (24.8) | 111 (25.4) | 1.11 [0.88–1.34] | |
| Sex | | | | |
| Female | 93 (85.3) | 364 (83.5) | 1 (ref) | |
| Male | 16 (14.7) | 72 (16.5) | 0.86 [0.70–1.04] | |
| **Contacts with colleagues** | | | | |
| Eating at the workplace canteen | | | | |
| Never/rarely | 63 (57.8) | 252 (57.8) | 1 (ref) | |
| Regularly/always | 46 (42.2) | 184 (42.2) | 1.00 [0.86–1.14] | |
| Breaks with other colleagues | | | | |
| Never/rarely | 44 (40.4) | 171 (39.2) | 1 (ref) | |
| Regularly/always | 65 (59.6) | 265 (60.8) | 0.95 [0.82–1.10] | |
| Change of outfit in workplace changing rooms | | | | |
| Never/rarely | 43 (39.4) | 225 (51.6) | 1 (ref) | 1 (ref) |
| Regularly/always | 66 (60.6) | 211 (48.4) | **1.92 [1.61–2.30]** | **1.93 [1.63–2.29]** |
| Participation in professional meetings | | | | |
| Never/rarely | 92 (84.4) | 349 (80.0) | 1 (ref) | 1 (ref) |
| Regularly/always | 17 (15.6) | 87 (20.0) | **0.73 [0.60–0.89]** | **0.72 [0.60–0.84]** |
| Airing of communal areas | | | | |
| Never/rarely | 28 (25.7) | 114 (26.1) | 1 (ref) | |
| Regularly/always | 81 (74.3) | 322 (73.9) | 1.03 [0.96–1.21] | |

OR: Odds ratio. ORa: Adjusted odds.

* Cases and controls were matched on workplace and occupation.

surgical masks and respirators against airborne viral infections [17]. Although this review was not specific to the SARS-CoV2 virus, current evidence suggests that surgical masks and respirators provide a similar level of protection, and respirators should be used selectively for greater risk situations such as aerosol-generating procedures due to the cost, discomfort and risk of badly-fitting respirators [18,19]. However, we cannot exclude that some HCWs cared for COVID-19 patients that were more critically ill, with longer hospitalizations, particularly in intensive care. In this case, the reduced risk of infection could have been due to less contagious patients, rather than to a protective effect of PPE.

When caring for non-COVID-19 patients, we found that HCWs who declared using gloves for all types of patient care had a higher risk of infection, possibly explained by improper use of PPE increasing risk of contamination. In this situation, HCWs may have considered risk of contamination to be very low. Vigilance to correct use of PPE may therefore be lowered, and risk of contamination due to misuse of PPE may increase. This has been proven for various infections, by decreasing the amount of hand hygiene HCWs perform when wearing gloves and cross contamination when HCWs do not systematically change gloves between patients [20]. Surprisingly, we found similar results for face shields and protective goggles, which HCWs however declared using very rarely in this situation. Again, improper use of these PPE or underuse of other PPE which should be associated with the face shields and goggles could

be an explanation. Wearing of protective overshoes was not found to be a protective measure, in accordance with guidelines [9,10].

We found an increase in risk of infection in HCWs who reported changing their uniform in a workplace changing room. Guidelines recommend wearing an outfit dedicated to the workplace as a protective measure [10,21], however communal changing rooms could increase spread of infection due to close proximity of HCWs to each other and removal of PPE during change of clothes. Before arriving on the ward and after leaving the ward, HCWs may consider the risk of contamination to be low and protective measures may therefore seem less important than when in contact with patients. The close proximity of HCWs in changing rooms due to same arrival and leaving times likely increases risk of transmission. HCWs who declared participating in meetings were found to be less at risk of infection. Due to establishment guidelines covering meeting rooms, HCWs were likely to wear proper PPE during meetings with other colleagues, thus decreasing risk of contamination. However there was no increased risk found with breaks with other colleagues or meals at the workplace canteen, two key moments when PPE, namely masks, are not worn. As with meeting rooms, health establishments produced guidelines and rules for these communal areas, with limits on the amount of people in break rooms and staff canteens, wearing of masks whenever possible and social distancing measures.

Airing of communal areas and of patients' or residents' rooms was not found to be a protective factor. Guidelines [22,23] recommend frequent room ventilation when possible in order to reduce potential airborne transmission [24,25]. The lack of association found with room airing evaluated by the question: "How often do you air communal areas and patients' or residents' rooms?" may be due to ventilation systems in place in most health establishments, therefore reducing effects of opening windows to air rooms. The frequencies may have lacked precision, with respondents choosing between 5 categories of frequency (never, rarely, regularly but less than every other day, regularly and more than every other day and every day). Guidelines [22,23] recommend airing rooms several times a day, therefore the question may have been too imprecise to provide an informative result.

This study presents some limitations. The questionnaire was filled in retrospectively by participants, and for respondents with history of COVID-19, the questions covered a short period of time before infection. HCWs were likely to not remember exactly what measures they applied during this period, although the infection was a noticeable event and had raised questions about its origin. This recall bias is unavoidable with studies based on questionnaires. Studies with a prospective measure of exposures and PPE use would help minimize this bias, observations of practices being more reliable than declarations, but are more difficult to conduct. Another limitation is that the information on the source of infection (workplace or community acquisition) was based on participants' declarations. Although contact tracing is performed by infection prevention teams for each case of COVID-19 in HCWs, the anonymous nature of the questionnaire did not allow us to verify the source of infection reported. Moreover, the use of an online questionnaire may have prevented certain profiles of caregivers from participating in the study. Another bias may result from an involuntary overestimation of PPE use by HCWs who declared a history of COVID-19 infection, explaining our results for PPE being a risk factor (namely respirators, face shields and protective goggles) during care of non COVID-19 positive patients. Another issue is that cases and controls who completed the questionnaire did so voluntarily. Therefore, respondents may not be fully representative of the general population of HCWs in France. Unfortunately, no estimation of the global response rate within HCWs in Normandy was able to be performed. The questionnaire was sent to health establishments' management and then relayed to HCWs. There was no feedback as to how many HCWs had access to the questionnaire from then on. Despite these biases, the fact

that our results are consistent with the data in the published literature and correspond to current recommendations allows us to believe that they are reliable.

Since the beginning of the pandemic, studies have demonstrated the higher risk of infection for HCWs, particularly exposed to the virus [11,26]. Until widespread vaccination, hand hygiene and correct use of PPE were the main barriers against the spread of COVID-19 infection [9,10,15,16,27]. Due to the emergence of new variants of the virus, with modifications of modes of transmission, infectivity, and response to vaccines, studying correct use of PPE is paramount [8]. Improper use of PPE should be highlighted as much as underuse of PPE. Because we are dealing with a moving target, further studies on risk factors and exposures are needed in order to minimize risk of infection within HCWs. Our results highlight the importance of proper use of PPE as a preventative measure against infection for HCWs.

## Supporting information

**S1 File.**
(CSV)

**S2 File.**
(CSV)

## Acknowledgments

We thank all of the participants in the study, as well as management and infection control teams who relayed the information about the study to health care workers in Normandy. All of the authors thank Josiane Lebeltel for her help in conveying the study to health establishments.

## Author Contributions

**Formal analysis:** Sarah Wilson, Pascal Thibon.

**Investigation:** Camille Jeanne-Leroyer, France Borgey, Emmanuelle Odinet-Raulin, Xavier Humbert, Simon Le Hello, Pascal Thibon.

**Methodology:** Audrey Mouet, Camille Jeanne-Leroyer, France Borgey, Simon Le Hello.

**Writing – original draft:** Sarah Wilson, Simon Le Hello, Pascal Thibon.

**Writing – review & editing:** Audrey Mouet, Camille Jeanne-Leroyer, France Borgey, Emmanuelle Odinet-Raulin, Xavier Humbert.

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
