## [Decision Letter · Decision Letter 0]

24 Jan 2022

PONE-D-21-34140Professional practice for COVID-19 risk reduction among health care workers : a Cross-Sectional Study with Matched Case-Control ComparisonPLOS ONE

Dear Dr. THIBON,

Thank you for submitting your manuscript to PLOS ONE. After careful consideration, we feel that it has merit but does not fully meet PLOS ONE’s publication criteria as it currently stands. Therefore, we invite you to submit a revised version of the manuscript that addresses the points raised during the review process.

Unfortunately, I was only able to secure one review. However, after carefully assessing the comments and the manuscript, I agree with the assessment of the reviewer. You describe a very interesting and relevant study. Nonetheless, your manuscript does require some modification before it can be considered for publication.

Specific comments are detailed below but, as you will see, the revisions required are, in general, very minor. I look forward to receiving your revised manuscript.

We look forward to receiving your revised manuscript.

Kind regards,

Ginny Moore

Academic Editor

PLOS ONE

Journal Requirements:

Additional Editor Comments:

Abstract (line 35) - please delete the figures 301 (14.6%) - it looks like these have been carried over from the results section. If word count allows, it would be helpful to clarify that the participants were HCW with/without contact with patients.

Introduction (line 61) - please amend citation(s) appropriately - should this read [2, 7] or [8]? Contamination of what? Do you mean "....against infection and/or onward transmission?"

Methods (line 74) - please clarify your study design. Was a single cross-sectional study and three case-control studies carried out? There is no mention of a cross-sectional study elsewhere in the manuscript.

Methods (line 95) - apologies, I was somewhat confused by the dates. The questionnaire was available from 29 March 21 to 30 June 21 but the study period for the case control studies was Sep 20 to 31 Jan 21. So were those with a history of COVID asked to provide the date of positive test and only those who tested positive between Sep and Jan were included as a case?

Table III - should the ORa associated with face shield also be in bold?

An increased risk of infection associated with PPE use with non-COVID patients is an interesting finding. Relatively few HCWs reported using mainly respirators and regularly using gowns - were they the same participants?

Table IV - an increased risk associated with changing in the workplace is another very interesting finding. Presumably, the data included in Tables II and III are associated with only those participants who are in contact with patients. Does Table IV include all participants regardless of profession?

Reviewers' comments:

Reviewer's Responses to Questions

**Comments to the Author**

1. Is the manuscript technically sound, and do the data support the conclusions?

Reviewer #1: Yes

2. Has the statistical analysis been performed appropriately and rigorously? 

Reviewer #1: Yes

3. Have the authors made all data underlying the findings in their manuscript fully available?

Reviewer #1: Yes

4. Is the manuscript presented in an intelligible fashion and written in standard English?

Reviewer #1: Yes

5. Review Comments to the Author

Reviewer #1: Dear authors,

The submitted study is well conducted and provides valuable information regarding HCW PPE use, measures to prevent inter-HCW spread of infection, and associated risk of acquiring covid-19 infection. Results are retrospective and rely on an internet based survey which confers some risks. Several of these are addressed adequatly in the discussison. However, I have a few points that I find would be of interest to clarify further in the discussion.

1. It is stated that the study was conducted during months when PPE was available and prior to vaccine availability. I think this period is a wize and deliberate choice. However, you state that most cases in 2020 were from March and April, at an early time point of the pandemic, when I suspect the situation was more chaotic. Were PPE supplies already sufficient by then so that recommended Infection control practices could be adhered to?

2. Even though a majority of covid-19 positive participants reported the possible source of infection was their work place, and most often infected patients, the uncertainty of such an assumption, specifically during wide spread dissemination in the community, should be discussed. Working with covid-19 patients will probably raise the suspicion of of them being the source, which is not necessarily true

3. Using respirators all the time or only when performing AGPs correlated with a reduced risk of infection. A reason could be better protection against contagious aerosols. An alternative protection could be that HCWs in these situations cared for patients that were more critically ill, at later time points of disease when viable virus less often can be recovered and the risk of infection therefore is significantly reduced. Working in intensive care with covid-19 patients has been associated with a reduced risk in previous reports.

The same may be true for the reduced risk associated with hair caps which may have been more commonly used by HCWs caring for critically ill patients (who are less contagious). Is this so, otherwize how can the reduced risk associated with hair cap use be reliably explained?

4. The study was performed prior to the alpha VOC, and I believe that the statement that covid-19 mainly spread via respiratory droplets in close proximity to an infectious person was true. Since then ever more infectious various variants have emerged from alpha via delta, and now omicron VOC. It needs to be discussed if the abundance of more infectious variants perhaps necessitates altered infection control measures, including PPE use, such as respirator use during care of covid-19 patients regardless of AGP. Perhaps also hint to an uncertainty as to whether the results of the same study, if performed today, would generate the same results. We are dealing with a moving target and infection control practices that were adequate yesterday may not be so today.

Kind regards

Ulf Karlsson, MD, PhD

6. PLOS authors have the option to publish the peer review history of their article (what does this mean?). If published, this will include your full peer review and any attached files.

Reviewer #1: **Yes: **Ulf Karlsson, MD

---

## [Author Response · Author response to Decision Letter 0]

1 Feb 2022

Professional practice for COVID-19 risk reduction among health care workers: a Cross-Sectional Study with Matched Case-Control Comparison

PONE-D-21-34140

>Thank you very much for considering our work. Please find below our responses to the comments of the academic editor and of the reviewer.

Editor:

1/ Abstract (line 35) - please delete the figures 301 (14.6%) - it looks like these have been carried over from the results section. If word count allows, it would be helpful to clarify that the participants were HCW with/without contact with patients.

>Agreed and done – The word count of the abstract is now 238.

2/ Introduction (line 61) - please amend citation(s) appropriately - should this read [2, 7] or [8]? Contamination of what? Do you mean "....against infection and/or onward transmission?"

>The right citation was [8]. We meant “onward transmission” and clarified this point in the sentence.

3/ Methods (line 74) - please clarify your study design. Was a single cross-sectional study and three case-control studies carried out? There is no mention of a cross-sectional study elsewhere in the manuscript.

>Thanks for this comment: we also think that our methodology needs to be clarified. In the questionnaire, we asked participants about their history of COVID-19 infection, vaccination status, and PPE applied at work. This is the “cross sectional” part of the study. We also collected in the questionnaire retrospective information on PPE use in case of history of COVID-19, which allowed us to carry out the case-control studies.

The mention “cross sectional study with matched case-control comparison” appeared in the title and in the “Material and method” part of the first version of the manuscript. We added mention to the “cross sectional” design in the section of the new manuscript presenting the results of this part of the study (line 145).

4/ Methods (line 95) - apologies, I was somewhat confused by the dates. The questionnaire was available from 29 March 21 to 30 June 21 but the study period for the case control studies was Sep 20 to 31 Jan 21. So were those with a history of COVID asked to provide the date of positive test and only those who tested positive between Sep and Jan were included as a case?

>Exactly! A total of 301 HCW declared a history of COVID-19. Of these 301, 109 had the infection during the case-control study period (sept 20->jan 21). We added “… history of COVID-19 infection with date of infection (confirmed by a positive SARS-CoV2 PCR or antigenic test” in the part “Period and Data acquisition” of the “Methods” section.

5/ Table III - should the ORa associated with face shield also be in bold?

>Agreed and done

6/ An increased risk of infection associated with PPE use with non-COVID patients is an interesting finding. Relatively few HCWs reported using mainly respirators and regularly using gowns - were they the same participants?

>They were not the same participants: of the 5 HCW declaring using mainly respirators, 3 declared using gowns regularly and 2 rarely.

7/ Table IV - an increased risk associated with changing in the workplace is another very interesting finding. Presumably, the data included in Tables II and III are associated with only those participants who are in contact with patients. Does Table IV include all participants regardless of profession?

>Table IV includes all participants, regardless of profession, with/without contact with patients. To clarify this point, we added “all professions included” in the title of table IV.

Reviewer #1:

The submitted study is well conducted and provides valuable information regarding HCW PPE use, measures to prevent inter-HCW spread of infection, and associated risk of acquiring covid-19 infection. Results are retrospective and rely on an internet based survey which confers some risks. Several of these are addressed adequatly in the discussison. However, I have a few points that I find would be of interest to clarify further in the discussion.

>Thank you very much for your comment. Please find our answers below. We agree with the limitations of the study related to the retrospective design and mode of data acquisition.

1. It is stated that the study was conducted during months when PPE was available and prior to vaccine availability. I think this period is a wize and deliberate choice. However, you state that most cases in 2020 were from March and April, at an early time point of the pandemic, when I suspect the situation was more chaotic. Were PPE supplies already sufficient by then so that recommended Infection control practices could be adhered to?

>The choice of the period for the case-control studies was indeed deliberate. At the early point of the pandemic, PPE supplies were not sufficient, and recommendations were still evolving. In response to a comment from the academic editor and to your comment, we clarified the fact that the cases included in the case-control studies were only those that occurred between September 2020 and January 2022, during a period when PPE was more readily available. Cases were asked to report their practice in the days before their infection.

2. Even though a majority of covid-19 positive participants reported the possible source of infection was their work place, and most often infected patients, the uncertainty of such an assumption, specifically during wide spread dissemination in the community, should be discussed. Working with covid-19 patients will probably raise the suspicion of of them being the source, which is not necessarily true

>In France (as in other countries most likely), cases of COVID-19 in healthcare workers are reported to infection control teams and contact tracing is performed. A search for sources of infection (community, other HCW, patient) is made, as well a screening tests among contacts. But we agree with you that the information on the source of infection was only based on declarations of participants. Since participation in the study was anonymous, it was impossible to verify the information.

We added this sentence in the Discussion part: “Another limitation is that the information on the source of infection (workplace or community acquisition) was based on participants’ declarations. Although contact tracing is performed by infection prevention teams for each case of COVID-19 in HCW, the anonymous nature of the questionnaire did not allow us to verify the source of infection reported.”

3. Using respirators all the time or only when performing AGPs correlated with a reduced risk of infection. A reason could be better protection against contagious aerosols. An alternative protection could be that HCWs in these situations cared for patients that were more critically ill, at later time points of disease when viable virus less often can be recovered and the risk of infection therefore is significantly reduced. Working in intensive care with covid-19 patients has been associated with a reduced risk in previous reports.

The same may be true for the reduced risk associated with hair caps which may have been more commonly used by HCWs caring for critically ill patients (who are less contagious). Is this so, otherwize how can the reduced risk associated with hair cap use be reliably explained?

>Thank you for this comment. We agree that the length of stay of patients could be a confounding factor in the link between the use of certain PPE and the reduction of risk: the reduction of risk being linked to less infectious patients rather than to the use of PPE. We did not address this point, which could be a convincing explanation for some of our findings (regarding hair caps especially).

We added this paragraph in the discussion: “However, we cannot exclude that some HCW cared for COVID-19 patients that were more critically ill, with longer hospitalizations, particularly in intensive care. In this case, the reduced risk of infection could have been due to less contagious patients, rather than to a protective effect of PPE.” 

4. The study was performed prior to the alpha VOC, and I believe that the statement that covid-19 mainly spread via respiratory droplets in close proximity to an infectious person was true. Since then ever more infectious various variants have emerged from alpha via delta, and now omicron VOC. It needs to be discussed if the abundance of more infectious variants perhaps necessitates altered infection control measures, including PPE use, such as respirator use during care of covid-19 patients regardless of AGP. Perhaps also hint to an uncertainty as to whether the results of the same study, if performed today, would generate the same results. We are dealing with a moving target and infection control practices that were adequate yesterday may not be so today.

>We totally agree with your comment, particularly in the current context of the diffusion of the omicron VOC. We modified our conclusion, and with your permission, we would like to use the term ”moving target”, which seems perfectly appropriate to us: “Due to the emergence of new variants of the virus, with modifications of modes of transmission, infectivity and response to vaccines, studying correct use of PPE is paramount [8]. Improper use of PPE should be highlighted as much as underuse of PPE. Because we are dealing with a moving target, further studies on risk factors and exposures are needed in order to minimize risk of infection within HCWs”

---

## [Decision Letter · Decision Letter 1]

7 Feb 2022

Professional practice for COVID-19 risk reduction among health care workers : a Cross-Sectional Study with Matched Case-Control Comparison

PONE-D-21-34140R1

Dear Dr. THIBON,

We’re pleased to inform you that your manuscript has been judged scientifically suitable for publication and will be formally accepted for publication once it meets all outstanding technical requirements.

Kind regards,

Ginny Moore

Academic Editor

PLOS ONE

Additional Editor Comments (optional):

Reviewers' comments:

Reviewer's Responses to Questions

**Comments to the Author**

1. If the authors have adequately addressed your comments raised in a previous round of review and you feel that this manuscript is now acceptable for publication, you may indicate that here to bypass the “Comments to the Author” section, enter your conflict of interest statement in the “Confidential to Editor” section, and submit your "Accept" recommendation.

Reviewer #1: All comments have been addressed

2. Is the manuscript technically sound, and do the data support the conclusions?

Reviewer #1: Yes

3. Has the statistical analysis been performed appropriately and rigorously? 

Reviewer #1: Yes

4. Have the authors made all data underlying the findings in their manuscript fully available?

Reviewer #1: Yes

5. Is the manuscript presented in an intelligible fashion and written in standard English?

Reviewer #1: Yes

6. Review Comments to the Author

Reviewer #1: (No Response)

7. PLOS authors have the option to publish the peer review history of their article (what does this mean?). If published, this will include your full peer review and any attached files.

Reviewer #1: **Yes: **Ulf Karlsson, MD, PhD, Dep of Infectious diseases and Infection Control, Skane University Hospital, Sweden

---

## [Editor Report · Acceptance letter]

10 Mar 2022

PONE-D-21-34140R1 

Professional practice for COVID-19 risk reduction among health care workers: a Cross-Sectional Study with Matched Case-Control Comparison 

Dear Dr. Thibon:

I'm pleased to inform you that your manuscript has been deemed suitable for publication in PLOS ONE. Congratulations! Your manuscript is now with our production department. 

Kind regards, 

on behalf of

Dr. Ginny Moore 

Academic Editor

PLOS ONE